# Cognitive Job Demands and Sports Participation among Young Workers: What Moderates the Relationship?

**DOI:** 10.3390/ijerph21020144

**Published:** 2024-01-28

**Authors:** Sara Wiertsema, Gerbert Kraaykamp, Debby Beckers

**Affiliations:** 1Radboud Social and Cultural Research, Department of Sociology, Interuniversity Centre for Social Science Theory and Methodology, Radboud University, 6525 XZ Nijmegen, The Netherlands; 2Behavioural Science Institute, Radboud University, 6525 XZ Nijmegen, The Netherlands

**Keywords:** physical activity, work stress, childcare, job autonomy, work–life balance

## Abstract

Cognitive job demands are theoretically and empirically associated with sports participation in various ways. Workers may be overwhelmed by stress and fatigue from their workload and therefore refrain from sports activities, but they can also feel the need to use sports as a way to recover and detach from work. The strategy to which workers adhere can depend on workers’ resources that moderate the cognitive job demands and sports participation relationship, such as educational attainment, being a parent, or having worktime and work location control. To test our expectations, we used recent information on sports participation by young working adults from the Netherlands (*N* = 2032). Using multinomial logistic regression modelling, we found that workers in mentally demanding jobs were more likely to participate in sports more than three times a week. In particular, workers without children reported a higher likelihood of participating in sports more than three times a week when they experienced high cognitive job demands. Among the higher-educated, workers with high cognitive job demands were less likely to participate in sports one to three times a week. We reflect on the academic and policy-related implications of our findings.

## 1. Introduction

European workers are increasingly working in occupations that are mentally demanding [1,2]. Some models forecast that 70 percent of Europe’s job growth in 2030 will be accounted for by mentally demanding jobs in the industries of health and social work; professional, scientific, and technical services; and education. The academic literature of the last decades chronicles the repercussions of being exposed to a chronically high mental workload [3,4,5,6]. For instance, the combination of a high mental workload and insufficient recovery opportunities has been linked to cardiovascular disease, type 2 diabetes, and burnout. In academic research, mentally tasking work is usually referred to as cognitive job demands, which are defined as the extent that work tasks require sustained mental effort by processing information, decision making, and learning [7,8]. Taking a closer look at cognitive job demands is important to sustain our understanding of the roles that excessive job demands play in the current labour force.

In addition to a shift in the labour market towards knowledge-based industries, we have also observed changes in the health of the labour force. Knowledge-based occupations are often characterised by physical inactivity and sedentary behaviour on the job. At the same time, people have become increasingly physically inactive during their leisure time, spending a large part of it sedentary. Accordingly, this decade’s prevalence of physical inactivity, sedentary behaviour, and unhealthy food intake are at alarmingly high levels [9,10]. Consequently, overweight and obesity levels have increased among European workers and so have the adverse health effects of overweight [11,12,13,14,15]. To counter this trend, people must be stimulated and facilitated to make healthy choices even when they are stressed and fatigued by their mental workload after a day at work. Physical activity and sports participation likely play a vital role in this process. Examples of sports participation include participating in basketball, fitness classes, or running, whereas leisurely walking and biking are considered general leisure-time physical activity. Sports participation is especially interesting because it arises from an explicit decision people make, and research shows that it can distinctly benefit one’s physical and mental health over spontaneous physical activities [16]. This raises a question regarding the extent to which work-related aspects, and especially cognitive job demands, relate to sports activity.

In the scientific literature, two competing hypotheses are mentioned regarding the relationship between cognitive job demands and sports participation. Workers may either let their cognitive workload negatively spill over into their leisure time *or* they may compensate for their cognitive load in their leisure time. The generalisation mechanism advocates that work-related experiences negatively spill over into the leisure domain [17,18,19]; in this perspective, stress and fatigue from cognitively demanding work would hinder workers from participating in sports. Various studies indeed found evidence for this association [20,21,22,23]. The opposing mechanism encompasses workers actively trying to compensate for cognitive load effects and/or unfulfilled needs of work in their leisure time [17,18,24,25]. Leisure-time sports participation then serves to replenish energy and help workers emotionally detach from work. A study by Mutz and colleagues [26] showed that time and performance pressure did not associate negatively with sports participation, and other studies reported similar outcomes [27,28]. Generalisation thus suggests that high cognitive demands associate with less sports participation, whereas compensation suggests the opposite. Previous studies strongly varied and found ambiguous results that supported both perspectives.

The ambiguity in theoretical and empirical understanding regarding the relationship between cognitive job demands and sports participation indicates that the direction is likely dissimilar for different groups of individuals; certain group features may moderate the association, making it either more positive or negative. Indeed, several overview studies have pointed to the importance of differentiating between social groups to understand differences in work-to-leisure behaviour [22,29]. We considered three potential moderating factors that cover three unique areas of workers’ lives: educational level (socioeconomic position), being a parent (family), and autonomy over worktime and work location (employment). Social, financial, cultural, or time resources associated with these moderators either strengthen the spillover of workload effects or enable compensation tactics. For instance, the higher-educated generally possess more resources than the lower-educated [30], so cognitive job demands may more positively relate to the sports participation of higher-educated workers. Workers with children more often experience time constraints [31], and in combination with cognitive job demands, this may exert additional pressure on sports participation. Finally, the well-known Job Demands–Resources model [32,33] proposed that workers’ autonomy provides resources to counter cognitive job demands that may replay positively to sports participation. This brings about our research question, namely: To what extent do workers’ educational attainment, being a parent, and having autonomy over worktime and work location moderate the relationship between cognitive job demands and sports participation?

By answering this research question, we aim to meaningfully add to the existing body of literature. Firstly, we progress the theoretical understanding of the cognitive job demands and sports participation relationship by expanding on the generalization and compensation mechanism with regard to sports participation. In addition, we formulate expectations on possible moderating influences that go beyond characteristics of work to create a better understanding of how workers’ daily lives influence and are influenced by work. Second, we employ unique and recent information on 2032 young Dutch workers to test our expectations [34]. Working young adults refers to workers between the ages of 18 and 34. Because research on work characteristics often includes the entire working population or is focused on middle-aged workers, e.g., [22,27,35], young adult workers received limited attention in the current research, and this raises the question whether our current knowledge is also suitable for this part of the labour force. Finally, the longitudinal structure of our data enabled us to relate cognitive job demands in 2021 to sports participation in 2022 and thus account to some extent for the causal order in looking at the association between cognitive job demands and sports participation.

## 2. Theoretical Background and Hypotheses

Leading theories on cognitive job demands and well-being largely originate from Herzberg [36], Hackman and Oldham [37,38], Karasek [33], and Siegrist [39]. More recently, the Job Demands–Resources theory has combined various perspectives and dominated the theoretical mindset of academic and policy-related occupational health researchers [7,32,40]. Most importantly, the various theoretical notions attest that work is an integral part of individuals’ and families’ daily lives. Work is not executed in a vacuum; workers have to divide their time and energy spent on paid labour as well as on activities such as chores, caregiving, leisure activities, and many others—with time spent in one life domain naturally coming at the expense of time spent in another [41,42]. Moreover, the amount of time allocated to paid labour and time schedules of paid labour are often fixed, which makes work inflexible compared to household chores and leisure-time activities [43]. As a result, work and its characteristics substantially impact people’s time and time use, including in sports participation.

Sports participation and its antecedents are also a much-studied phenomenon. Not only the amount of participation but also the location, type of sports, and with whom one participates in sports have been intensely studied, e.g., [44,45]. Regarding the relationship with job characteristics, the focus of research is often on (leisure-time) physical activity or exercise. However, sports participation has distinct characteristics as it (specifically in European countries) often refers to competition and community sports in which governments play a large role [46]. Studying sports participation captures the conscious decision by people to be active, and it is a health behaviour that is closely linked with governmental policy. Consequently, understanding the relationship between cognitively demanding work and sports participation is vital to individual workers and to policymakers who want to maintain a healthy workforce.

In the following sections, we elaborate on two mechanisms through which cognitively demanding work relates to sports participation, namely generalization and compensation. We further elaborate on whether this relationship could be different for different groups. More specifically, we focus on the moderation of this relationship by educational level, being a parent or not, and the level of autonomy over worktime and work location.

### 2.1. The Generalisation Mechanism

The notion of generalisation implies that there is a spillover between events and emotions at work and outside of work [17,18,19]. Workers who experience high levels of job demands are, according to this mechanism, also experiencing its consequences outside of work in their leisure time. The main consequence of cognitively demanding work is mental strain, which causes people to feel (mentally) fatigued and/or stressed [47,48]. It further means that after work they need to recover; for instance, through media consumption (i.e., watching television, social media, or reading) or sleep [49]. Fatigued or stressed workers often feel overloaded, lack energy, and lose concentration easily, and therefore they may refrain from energy-consuming leisure-time activities such as sports [50]. This also corresponds with the notion that stressed individuals would prefer to cope with work overload through risky and unhealthy behaviours instead of healthy ones [51,52]. According to this perspective, workers with high cognitive demands will thus lack the energy and are too tired to participate in sports.

### 2.2. The Compensation Mechanism

Contrary to a negative spillover, empirical evidence sometimes points to a positive association between cognitive job demands and sports participation. This perspective is referred to as compensation, and it expects that workers may use sports to recover and compensate for work-related stress and fatigue. Recovery is seen as essential after a day of mentally demanding work, and people can realize this through detaching from work and through positive experiences; for instance, by being physically active in sports [48]. In other words, workers with high cognitive demands may actively compensate through sports to regain energy and detach from work [17,25,53,54]. Prior research supports this claim by showing that often-mentioned reasons to participate in sports are reducing stress, having fun, and increasing energy levels [55,56]. For example, a worker with high mental job demands could attend a yoga class or go for a run because this will help with detaching from work and regaining energy.

### 2.3. Moderation of the Cognitive Job Demands and Sports Participation Relationship

Expectations that have been based on the generalization or compensation mechanism regarding how cognitive job demands relate to sports participation contradict each other. One way of understanding these opposing expectations is by examining whether generalisation or compensation mechanisms are more likely to be prevalent among certain social groups. Analytically, we presume that the association between cognitive job demands and sports participation is moderated because workers possess specific resources or experience specific restrictions. We here rely on a neo-Weberian theoretical framework in which individuals are presumed to have agency and make choices dealing with personal resources or restrictions. More specifically, we expect here that the relationship between job demands and sports participation differs according to a worker’s educational level, a worker’s level of autonomy over worktime and work location, and because of the restrictions of being a parent. These conditions mean that workers are differently equipped to deal with the strain of cognitive job demands in deciding to participate in sports.

First, having an extensive education is generally perceived as an indication that people possess more information and knowledge on the positive sides of sports participation [57]. The higher-educated more often recognize the impact of sports participation on their health, hold a more resourceful and sporting social network, and have financial opportunities to cover the costs of sports regarding clothing, contributions, and travel than their lower-educated counterparts [30,58,59]. This means that for the higher-educated, it is easier and more apparent to counteract work overload or stress through sportive activity. Contrarily, the lower-educated are less equipped and generally lack resources to prevent their cognitive workload from hindering their sports participation. We therefore expect that (H1):

**Hypothesis** **1.**
*The association between cognitive job demands and sports participation is more positive or less negative for higher-educated workers than for lower-educated workers.*


Second, workers’ amount of leisure time is not only determined by their working hours but also by other forms of restricted or committed time [29,41,43]. It is generally acknowledged that family obligations, such as caring for family members, are relatively time-consuming and ask a lot from people [31]. Snyder’s [60] multiple loads theory suggests that individuals with multiple roles, such as worker and caregiver, endure strain due to limits in time and energy [58]. Coming home fatigued and stressed due to mentally demanding work and encountering many kinds of family obligations (making dinner or bringing children to friends or clubs) may intensify strain and stress and leaves less time to participate in sports. So, it is likely that especially high-demand workers who have children are hindered in participating in sports due to their childcare responsibilities [31,43]. We therefore hypothesize that (H2):

**Hypothesis** **2.**
*The association between cognitive job demands and sports participation is less positive or more negative for workers with children than for workers without children.*


Third, various versions of job demand theories refer to resources on the job that may reduce the negative spillover of mental workload, e.g., [32,33]. They imply that certain job features (job resources) can alleviate cognitive, emotional, or physical burdens at work (job demands) and help a worker to achieve work goals, reduce demands, and stimulate personal growth. Recent research and reviews regarding the Job Demands–Resources model showed its importance across different contexts, work characteristics, and outcomes e.g., [61,62]. With respect to the relationship between cognitive job demands and sports participation, we assume that having job resources such as autonomy or control over worktime and work location may be essential to reduce workload and facilitate leisure-time sports participation. Workers who can influence when to start and finish work, have flexibility in scheduling tasks and breaks, and have the opportunity to work from home are better equipped to combine work with life [49,63,64]. In other words, having worktime and work location control offers more flexible options to plan leisure time, as it relaxes temporal and areal restrictions, which makes participation in sports easier and facilitates compensation for workload through sports [63,64,65]. This suggests that flexibility in worktime and work location mitigates the consequences of cognitive job demand overload, which may give workers more opportunities and energy to participate in sports [65,66]. For this reason, we expect that (H3):

**Hypothesis** **3.**
*The association between cognitive job demands and sports participation is more positive or less negative for workers with more autonomy over worktime and work location than for workers with less autonomy over worktime and work location.*


## 3. Materials and Methods

To test our expectations, we employed information from the TRansition Into Active Living (TRIAL) data collection [34]. The TRIAL data consisted of a sample of people aged 16 to 40 years old from the I&O Research Panel [67]. Individuals in the panel were selected through random samples of (municipal) population registers and comprehensive address files. Panel members received points when completing a survey that could be exchanged for gift cards. The survey focused on life course transitions, physical (in)activity, and their causes and consequences. In the first round of the TRIAL data collection (conducted in October 2021), 4691 respondents aged 16 to 40 responded to the online questionnaire (response: 46%), and in the second round (October 2022), 3206 respondents participated (panel attrition: 32%). For our study, we selected respondents who were paid workers for at least 20 h a week in both 2021 and 2022 (*N* = 2059; excluded: 36%). We further included workers with valid responses on the sports participation question in 2022 and on cognitive job demands, educational attainment, household composition, worktime and work location, and the control variables in 2021 to ensure an equal number of cases across our models (excluded: 1.4%). Our selection led to an analytical sample of 2032 workers aged between 18 and 39. We weighed the sample according to sex, age, education, and region.

### 3.1. Measurements

Our dependent variable *sports participation* was derived from the item ‘In the past month (September), have you participated in sports? If yes, which sports did you do?’. For each sport, respondents indicated the frequency of participating in this sport: daily, 3 to 5 times a week, 1 to 2 times a week, and 1 to 3 times a month. We recoded values for all sports (max. 3) to weekly sports intensity (i.e., 7, 4, 1.5, or 0.5) and summed the intensity for different sports to arrive at a measure for total weekly sports participation (range: 0 to 15.5). As reported in all sports research, the frequency of sports participation was not normally distributed but right-skewed (see Appendix A). We therefore decided to construct three nominal categories that mirror the distribution of sports events: no weekly sports (39%), 1 to 3 weekly sports events (37%), and more than 3 weekly sports events (24%). We checked our analysis with alternative operationalisations, but this did not affect the main results of our study.

Workers’ *cognitive job demands* were indicated by a list of five items. These and the items regarding job autonomy over worktime and work location were derived from Karasek’s Job Content Questionnaire and their Dutch application in the Netherlands Working Conditions Survey [68,69]. Respondents were asked to what extent they experienced high cognitive functioning, exhaustion at the end of the day, enough time to finish tasks, working quickly, and work causing stress, with answer options ranging from (almost) never to (almost) always. Confirmatory factor analysis and reliability analysis indicated that the five items represented one underlying dimension (Cronbach’s α = 0.680). To construct a scale, we averaged the scores on the five items.

We included three moderating factors in our analyses. *Educational attainment* was measured using ISCED levels of the Dutch educational system (see Appendix A). We recoded ISCED levels to years of education ranging from 4 (unfinished primary school) to 16.5 (finished university master). Note that the average number of years of education is relatively high in our sample. This is plausible, however, since the sample was restricted to workers between 18 and 39 years of age who worked at least 20 h a week. *Being a parent* was measured with the question ‘Do you have children?’ and was categorized as being a parent (1) or not being a parent (0). We constructed a measure of *control over worktime and work location* based on two items, one on control over start and finishing worktime and one on the opportunity to work from home (Cronbach’s α = 0.758). We averaged scores (ranging from (almost) never to (almost) always) to calculate a measure that represented control over worktime and work location.

We controlled in our models for several individual and work characteristics because variation in cognitively demanding work and sports participation has been attributed to such characteristics [26,27,35]. Workers’ sex was a dummy variable, coded *male* (1) and female (0). *Age* originally ranged from 18 to 39 but for ease of interpretation, we recoded age to have a meaningful zero (0 to 21). We also included whether workers were sometimes, often, or (almost) always *hampered by health* (1) in the past month ((almost) never = 0). We used a question on *cohabiting with a partner* as a dichotomous variable that indicated whether a respondent cohabited with a partner (1) or not (0).

We also controlled for several working-related aspects. First, we included contracted *working hours*, which originally ranged from 20 to 50 h per week (bottom coded 0 = 20). Also, whether respondents were employed under a temporary (0) or *permanent contract* (1) was indicated by a dichotomous variable. Lastly, workers’ *physical labour* was measured with items on using force, lifting, pushing, pulling or dragging, and standing or walking. We averaged scores to arrive at a scale on which a high score indicated more intense physical labour (Cronbach’s α = 0.720). Table 1 displays descriptive statistics according to category of weekly sports participation (correlations are provided in Appendix A).

### 3.2. Analytical Strategy

Since weekly sports frequency was a nominal variable with three categories—no weekly sports participation, one to three weekly sports events, and more than three weekly sports events—we employed multinomial logistic regression with Stata 17.0 (mlogit) to test our hypotheses. More specifically, we explored the likelihood of participating one to three times a week or more than three times a week in sports compared to not participating at all in relation to the independent variables. We utilized multinomial logistic regression because we assumed that the distances between the categories was not equal. Attitudes and behaviours that affect participating in sports one to three times a week may be different than participating more than three times. The results are presented in two tables. Table 2 comprises Model A, in which we tested the direct, unadjusted effect of cognitive job demands on sports participation; Model B, which includes all control variables; and Model C, which additionally displays the direct effects of the moderators. In Table 3, moderation was tested by including the interaction effects separately in Models 4a, 4b, and 4c. Probabilities based on the estimates were calculated utilizing the following formula: $$P=\frac{1}{1+e^{-(B0+B1\times 1+\ldots +BnXn)}}$$

## 4. Results

The results in Table 2 show that with regard to one to three weekly sports events, cognitive job demands were not related to a higher or lower probability of participating in sports on a weekly basis (b = 0.055, SE = 0.102). In contrast, cognitive job demands were positively related to the likelihood of participating in sports more than three times a week compared to not participating at all (b = 0.280, SE = 0.113). This indicated that workers with high mental work demands seemed to compensate for their job stress by being physically active in sports more often. Workers with the highest job demands were approximately twenty percent more likely to participate in sports more than three times a week compared to workers with the lowest job demands (*p* = 0.49 to *p* = 0.29). Model 2 shows that this result was not affected by including our control variables. These variables show the expected effects: older workers, people with a partner, those hampered by health issues, and those performing physical labour had a lower chance to participate in sports. Remarkably, workers working longer hours were slightly more active in sports, which might indicate a healthy worker effect.

In Model 3, we observed that the higher-educated (b = 0.128, SE = 0.029), workers without children (b = −0.352, SE = 0.166), and workers with more control over worktime and location (b = 0.157, SE = 0.070) had a higher chance to participate in sports three times or more a week than to not participate in sports. In this model, the earlier-mentioned effect of cognitive job demands was no longer significant. This signals that educational attainment, being a parent, and work control induced shifts in the relationship between cognitive job demands and sports participation. This warranted our approach of looking at moderation and including interaction effects in Table 3.

In Table 3, a significant difference in the association between cognitive job demands and sports participation was observed for workers with different levels of educational attainment (also see Figure 1). This negative interaction effect indicated that the positive effect of cognitive job demands on the chance to participate in sports one to three times a week weakened when a worker was higher-educated (b = −0.102, SE = 0.046). For lower-educated workers, higher cognitive job demands still related to higher probabilities of being active one to three times a week compared to not being active in sports. In particular, the highest-educated workers were no longer able to compensate for their mentally demanding work, and their cognitive job demands led to lower sports participation; in this group, mentally demanding work was associated with a lower likelihood of having one to three weekly sports events. Interestingly, the probability of participating in sports more than three times a week was not affected by the moderation of years of education. There thus seemed to be a substantial difference between higher-educated workers who were active in sports a few times a week and those who were very frequently active during the week. Although our results are interesting, we have to reject Hypothesis 1.

Our second hypothesis reflected the expectations of the moderating effect of being a parent (Figure 1). The negative interaction estimated in Model 4c (Table 3) indicates that the positive effect of cognitive job demands on the chance to participate in sports more than three times a week became weaker when workers had children (b = −0.772, SE = 0.288). Workers without children with high cognitive job demands still had a high probability of being active more than three times a week compared to not being active at all. Workers with children no longer seemed to be able to perform sports regularly, since in this group mentally demanding work was associated with a lower likelihood of having more than three weekly sports events. This result confirmed Hypothesis 2.

Table 3 also shows that control over worktime and work location did not moderate the association between cognitive demands and moderate or intensive sports participation. We therefore reject Hypothesis 3. In relation to earlier findings within the job demands–control framework, this is an important finding when considering our sample of young workers—the effect of cognitive job demands on sports participation did not seem to be dependent on autonomy over worktime and work location.

We performed additional robustness analyses to check the stability of our results (see Appendix A). First, an ordinary least squares regression on linear weekly sports frequency (interval level) reflected our previous results regarding being a parent and worktime and work location autonomy. We no longer found a significant moderation by workers’ educational level. This actually supported our initial choice to employ a categorisation of sports participation. By using an interval measure of sports participation, important information was hidden in the assumption of linearity. We secondly examined results of multinomial logistic regression analyses separately for male and female workers. From this, we learned that the negative interaction effect of education was primarily driven by female workers—the positive effect of cognitive job demands on the chance to participate in sports became weaker as a female worker was more highly educated. For male workers, especially being a parent strongly and negatively affected the positive relationship between cognitive job demands and frequent sports participation.

## 5. Conclusions and Discussion

Although many studies from sociology and labour psychology have explored relationships between work characteristics, family load, and education on the one hand and leisure-time physical activity/sports participation on the other, their combined and moderating influences had so far received surprisingly limited scientific attention. Additionally, ambiguous and varying outcomes pointed to a complex relationship between work, stress, and sports participation. As a step towards filling this gap in the scientific literature, we examined whether specific groups of workers possessed resources to compensate for their workload effects by participating in sports and whether others lacked such resources and therefore experienced more general daily life consequences of cognitive work stress. Specifically, we studied the extent to which cognitive job demands affect an individual’s sports participation as well as the extent to which this influence varied across social groups exemplified by educational attainment, childcare responsibilities, and worktime and work location autonomy.

Regarding the first issue, our findings indicated that cognitive job demands generally were positively associated with higher numbers of weekly sports events in our representative sample of young Dutch workers. This is largely in line with the theoretical notion of compensation and indicates that when personal recovery needs are strong enough, they can compel a cognitively challenged worker to participate in sports. In line with earlier works [17,25,53,54], we thus confirm that workers with high cognitive work demands can actively compensate through sports to regain energy and detach from work. This result clearly contradicts the inverse notion of the generalisation of cognitive work demands [22]. This anomaly may be explained by the fact that consequences of cognitive job demands are almost exclusively studied in the organizational psychological literature, in which the main focus is on a broader concept of leisure-time physical activity. As advocated in the Introduction, sports participation stands out from general physical activity because it is an explicit decision people make, and research shows that it distinctly benefits one’s physical and mental health over spontaneous physical activities [16]. We therefore advocate that our study may be taken as a starting point to further investigate the association between cognitive job demands and sports participation in representative national samples and with longitudinal designs.

In answering our second research question, it was revealed that the association between cognitive job demands and sports participation indeed varied between specific social groups. First, it was established that cognitive job demands were associated with a lower chance to participate frequently in sports for workers with children, whereas for workers without children, these job demands related to a higher chance to participate in sports frequently. So, workers with children may want to compensate for their cognitive job demands, but because of their childcare responsibilities, they are unable to do so. This result confirms earlier findings on the impact of multiple roles and the toll this can take on workers [58,60].

Second, we found that even though the highly educated were more likely to be active in sports, when this was combined with high job demands, their workload seemed too demanding to be active in sports. This finding is in contrast with our expectation that because of their abundant informational, cognitive, and social resources, the higher-educated would be better able to compensate for mentally demanding work through sports [30,57,58]. An explanation for our finding lies in the subjectivity and expectations regarding cognitive job demands. The higher-educated are specifically trained and prepared for jobs that hold cognitive job demands. The experience of working in a job with high job demands and its objective influence may thus be different for workers with distinct educational backgrounds. Our results interestingly show that years of education does not matter for the relationship between cognitive job demands and sports participation among those participating in sports more than three times a week. This group of sports aficionados probably hold additional motives or habits that make them more active in sports in the light of a heavy workload [55,70]. This also warrants the categorical distinction we make in sports behaviour between workers without, with moderate, and with frequent sports participation.

Third, our results displayed no differentiation in sports participation between workers with or without control over worktime and work location. This finding is in contrast with expectations from the Job Demands–Resources model, in which relevant job resources such as autonomy are expected to moderate the roles of job demands [32]. This irregular finding can potentially be explained by our sample of relatively young workers. Young workers are generally characterized by having more stress responses to cognitive job demands, less working experience, and a higher focus on leisure than on work [71,72,73]. Subsequently, young workers respond with stress to high job demands, but as their focus is already more directed towards leisure time, additional work autonomy may potentially not make much of a difference in alleviating work stress.

Surely, our study has limitations as well as suggestions for future research. First, our general approach is characterized by its sports causation perspective (i.e., individuals’ experiences at work impact subsequent sports participation). We are, however, aware that reverse causation is imaginable. This would mean that workers’ sports participation and possible health gain would lead to more success in occupational life with better and cognitively challenging jobs [74,75]. In our study, it was not possible to deal with this so-called sport selection, but we attempted to control for possible confounders to the best of our ability (i.e., health and physical work). As a second drawback, it should be clear that in our study, we employed self-reported measures of sports participation. From previous research, it is clear that individuals likely overestimate their physical activity [76]. However, to influence our results, this bias had to be socially selective, and there was no clear indication of that in our study. Future research may profit more from more elaborate measures (i.e., trackers and diaries) to indicate an individual’s weekly or daily sporting activities. Moreover, the TRIAL data does not include information on the duration of workers’ sports participation. Respondents who participated in sports once a week could thus play sports for 20 min or for 3 h and were still assigned to the same category. However, monthly and weekly sports frequency is the leading indicator of sports participation used by the Dutch government [77]. To this end, our study complied with this framework. Finally, because our conclusions were largely based on cross-sectional data, we did not attempt to make causal claims and focused on relationships and associations. Future research may want to employ longitudinal methods to establish the consequences of “changes” in cognitive job demands for workers’ sports participation.

To end, we want to discuss some of the implications of our study. This study attempted to advance theoretical rigor by investigating the puzzling work–leisure relationship to uncover dividing lines between social groups in dealing with cognitive work demands through sports participation. First, a novel perspective might be that young workers nowadays react differently to mechanisms and concepts that have been important for a long time. The rise of working from home [65], burnout, and lower well-being [78] as well as the centrality of leisure instead of work among these young adults [73] calls for innovative research on the consequences of working conditions among this group. These young workers will constitute the labour force for the next decennia, and thus suitable knowledge regarding their behaviour and preferences is vital to create and maintain a healthy workforce. Second, for young workers, sports participation seems to be a way to relieve stress and fatigue from cognitively challenging work. This seems especially true for workers without childcare responsibilities and for lower-educated workers. In contrast, for higher-educated workers and parents, high cognitive demands on the job are associated with lower chances of being active in sports. Therefore, policies or interventions could be geared towards this group to offer opportunities to maintain a healthy lifestyle, and companies may want to facilitate sports by interlacing sports and physical activity within their organisations [79,80].

## Figures and Tables

**Figure 1 ijerph-21-00144-f001:**
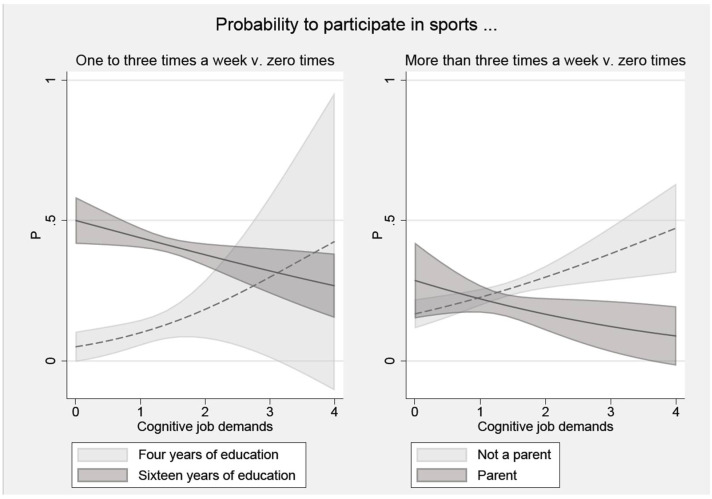
Visualization of the moderation effect of educational attainment and being a parent. Note: the figure is based on Models 4a and 4b. All other variables in the models are included with the average across our sample.

**Table 1 ijerph-21-00144-t001:** Descriptive statistics according to categories of weekly sports participation.

					Categories of Weekly Sports Frequency					
			Total		None (*N* = 794)		One to Three (*N* = 753)		More Than Three (*N* = 485)	
	Range		Mean	S.D.	Mean	S.D.	Mean	S.D.	Mean	S.D.
Cognitive job demands	0	3	1.352	0.504	1.352	0.512	1.360	0.495	1.423	0.500
Years of education (6 years = 0)	0	10.5	8.548	2.336	7.865	2.595	9.069	1.933	8.9	2.151
Being a parent (=1)	0	1	0.285		0.338		0.285		0.197	
Control over worktime and work location	0	3	1.482	1.055	1.281	1.051	1.596	1.046	1.646	1.024
Male (=1)	0	1	0.558		0.555		0.530		0.606	
Age (18 years old = 0)	0	21	13.204	4.702	13.749	4.71	12.939	4.650	12.699	4.685
Hampered by health (=1)	0	1	0.276		0.296		0.284		0.230	
Cohabiting with a partner (=1)	0	1	0.574		0.610		0.600		0.478	
Working hours (20 h = 0)	0	30	17.061	6.994	16.742	7.440	16.646	6.597	18.206	6.692
Permanent contract (=1)	0	1	0.707		0.721		0.693		0.703	
Physical labour	0	3	0.793	0.749	0.939	0.791	0.704	0.687	0.682	0.727

*Source*: TRansition Into Active Living (2021 and 2022); *N* = 2032.

**Table 2 ijerph-21-00144-t002:** Multinomial logistic regression on sports participation (reference category is ‘no weekly sports events’).

	Model 1			Model 2			Model 3		
	b		SE	b		SE	b		SE
**One to three weekly sports events**									
Cognitive job demands	0.055		0.102	0.055		0.108	−0.063		0.111
Years of education							0.201	***	0.027
Being a parent (no = ref.)							−0.132		0.140
Control over worktime and work location							0.102		0.062
Male (female = ref.)				−0.106		0.112	−0.058		0.114
Age				−0.041	***	0.012	−0.036	**	0.013
Hampered by health (no = ref.)				−0.060		0.116	0.020		0.118
Cohabiting with a partner (no = ref.)				0.023		0.109	0.029		0.121
Working hours				−0.003		0.008	−0.009		0.008
Permanent contract (no = ref.)				−0.050		0.117	0.033		0.119
Physical labour				−0.437	***	0.070	−0.102		0.090
Intercept	−0.181		0.148	0.864	***	0.251	−1.134	**	0.355
**More than three weekly sports events**									
Cognitive job demands	0.280	*	0.113	0.280	*	0.121	0.197		0.123
Years of education							0.128	***	0.029
Being a parent (no = ref.)							−0.352	*	0.166
Control over worktime and work location							0.157	*	0.070
Male (female = ref.)				0.086		0.127	0.116		0.129
Age				−0.048	***	0.013	−0.037	**	0.014
Hampered by health (no = ref.)				−0.348	*	0.136	−0.294	*	0.138
Cohabiting with a partner (no = ref.)				−0.437	***	0.120	−0.352	**	0.133
Working hours				0.020	*	0.009	0.012		0.010
Permanent contract (no = ref.)				0.021		0.133	0.084		0.134
Physical labour				−0.453	***	0.081	−0.150		0.102
Intercept	−0.890	***	0.168	0.026		0.283	−1.453	***	0.397

*Source:* TRansition Into Active Living (2021 and 2022); *N* = 2032. * *p* < 0.05; ** *p* < 0.01; *** *p* < 0.001 (two-tailed).

**Table 3 ijerph-21-00144-t003:** Moderation analyses from multinomial logistic regression on sports participation (reference category is ‘no weekly sports events’).

	Model 4a			Model 4b			Model 4c		
	b		SE	b		SE	b		SE
**One to three weekly sports events**									
Cognitive job demands	0.814	*	0.412	−0.027		0.131	−0.031		0.185
Years of education	0.335	***	0.067	0.200	***	0.027	0.200	***	0.027
Being a parent (no = ref.)	−0.127		0.140	0.006		0.349	−0.131		0.140
Control over worktime and work location	0.092		0.063	0.103		0.062	0.136		0.148
Cognitive job demands *									
Years of education	−0.102	*	0.046						
Being a parent (no = ref.)				−0.103		0.235			
Control over worktime and work location							−0.026		0.101
Intercept	−2.226	***	0.615	−1.179	**	0.364	−1.170	**	0.392
**More than three weekly sports events**									
Cognitive job demands	0.606		0.448	0.362	**	0.140	0.334		0.209
Years of education	0.195	**	0.074	0.128	***	0.029	0.125	***	0.030
Being a parent (no = ref.)	−0.347	*	0.166	0.695		0.417	−0.346	*	0.166
Control over worktime and work location	0.151	*	0.071	0.165	*	0.071	0.282		0.168
Cognitive job demands *									
Years of education	−0.051		0.050						
Being a parent (no = ref.)				−0.772	**	0.288			
Control over worktime and work location							−0.091		0.112
Intercept	−1.959	**	0.680	−1.674	***	0.407	−1.612	***	0.443

*Source:* TRansition Into Active Living (2021 and 2022); *N* = 2032. * *p* < 0.05; ** *p* < 0.01; *** *p* < 0.001 (two-tailed). Note: all models include the control variables from Model 3 in Table 2.

## Data Availability

The data are not publicly available at the current moment due to ongoing data collection. At a later moment, the data will be publicly available. More information is available upon request from the corresponding author.

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
