# Peer review of "Cognitive Job Demands and Sports Participation among Young Workers: What Moderates the Relationship?"

_ijerph, 2024, doi:10.3390/ijerph21020144_

Round 1

Reviewer 1 Report

Comments and Suggestions for Authors

Thank you for the opportunity to review the manuscript titled "The Interplay Between Cognitive Job Demands and Sports Participation Among Young Workers: What Moderates the Relationship?" The study focuses on investigating the moderator between cognitive job demands and sports participation. Overall, it presents an interesting and rigorously designed study. I appreciate the authors' efforts in conducting this work and providing meaningful implications in this regard. I have a few suggestions that I hope will be helpful for your revisions:

1.     In the introduction, you mention, "To study this moderation controversy, we consider three potential moderating factors among workers: educational level, being a parent, and having autonomy over work time and work location." I suggest elaborating more on the reasons for selecting these three variables as potential moderators. What is the logic behind this choice, and what references support it?

2.     Also, in the introduction, you state, "These are often portrayed as ‘workers of the future,’ but little research focuses specifically on this group." I am curious about the focus of most studies in this area. Which age groups attract the most attention? Perhaps a brief introduction to this would help readers better understand your study.

3.     Regarding participants, more clarity and detail about your sampling process is suggested. First, was there any compensation provided to the participants? Also, consider using a table or figure to illustrate your inclusion and exclusion process.

4.     In the method section, please specify how you designed the items for measuring workers' cognitive job demands.

5.     Finally, for the analysis strategy, please specify the rationale behind the linearity of each model. What are the reasons for setting up the models in this particular way?

Reviewer 2 Report

Comments and Suggestions for Authors

Dear Authors,

Thank you for your contribution, which I found very interesting.

The paper is well organized and written. I have only one concern about the references. Please find more recent references (for example, JD-R model, you can find articles in 2023). 

Furthermore, I suggest to consider different studies based on JD-R model, to demonstrate that the theoretical framework is valid in different fields and applications. Extend the theoretical introduction with a subparagraph on this.

Comments on the Quality of English Language

I think that an overall check of the English must be done. 

Reviewer 3 Report

Comments and Suggestions for Authors

The authors of the article raised the fundamentally important issue of health conservation, establishing a link between cognitive requirements for work and sports. The authors raise the most important questions regarding the compensation of stress and fatigue from cognitively difficult work by means of sports. Meanwhile, the article would have been presented more qualitatively if the authors had paid attention to some issues.

In the "methods" section, it is necessary to highlight the methods of data processing – using which software the authors established the significance of differences, regression coefficients and moderation effects. In discussing the results, some confusion in concepts is found: on the one hand, the authors include as a variable and discuss participation in sports (occupation), on the other – in sports (competition), finally, the authors contrast this with physical activity in their free time. In this case, the question arises: for example, does going to the pool 3 times a week relate to sports or physical activity in your free time? Or is it essential for the authors to single out an occupation in competitive sports? Another limitation of the study is the inability to control the duration of sports activities, since the question is about activities in the last month, and not regular sports.  It would be useful to highlight the main results of the study more clearly in conclusion: what scientific and theoretical knowledge was obtained as a result of the study.

I wish success to the authors in submitting the article to the journal.
